# Molecular Detection and Phylogenetic Relationships of Honey Bee-Associated Viruses in Bee Products

**DOI:** 10.3390/vetsci11080369

**Published:** 2024-08-12

**Authors:** Delka Salkova, Ralitsa Balkanska, Rositsa Shumkova, Stela Lazarova, Georgi Radoslavov, Peter Hristov

**Affiliations:** 1Department of Experimental Parasitology, Institute of Experimental Morphology, Pathology and Anthropology with Museum, Bulgarian Academy of Sciences, 1113 Sofia, Bulgaria; dsalkova@abv.bg; 2Department “Special Branches”, Institute of Animal Science, Kostinbrod, Agricultural Academy, 1113 Sofia, Bulgaria; r.balkanska@gmail.com; 3Research Centre of Stockbreeding and Agriculture, Agricultural Academy, 4700 Smolyan, Bulgaria; rositsa6z@abv.bg; 4Department of Animal Diversity and Resources, Institute of Biodiversity and Ecosystem Research, Bulgarian Academy of Sciences, 1113 Sofia, Bulgaria; stela.lazarova@gmail.com (S.L.); gradoslavov@gmail.com (G.R.)

**Keywords:** honey bee-associated viruses, RT-PCR, phylogeny, honey bee products

## Abstract

**Simple Summary:**

One of the major threats to the health status of honey bee colonies is presented by honey bee-associated viruses. Therefore, the detection and identification of viruses in bee colonies is an important step in the fight against them. In this study, we used a novel and non-invasive approach—molecular analysis of environmental RNA as a tool for detection of six of the most widespread viruses in honey bee colonies in bee products (pollen, bee bread, and royal jelly) from different regions in Bulgaria. The obtained results showed successful detection and identification of the Deformed wing virus, the Acute bee paralysis virus, the Black queen cell virus, and the Israeli acute paralysis virus in all the investigated bee products. Phylogenetic analysis of the detected Bulgarian isolates was implemented to reveal the phylogenetic relationships with the highly similar worldwide strains available in the GenBank database. Ultimately, honey bee products represent a valuable source of eRNA, which allows the monitoring of virus infections in bee colonies at local, regional, and even national levels.

**Abstract:**

In the last few years, the isolation and amplification of DNA or RNA from the environment (eDNA/eRNA) has proven to be an alternative and non-invasive approach for molecular identification of pathogens and pests in beekeeping. We have recently demonstrated that bee pollen and bee bread represent suitable biological material for the molecular identification of viral RNA. In the present study, we extracted total RNA from different bee products (pollen, n = 25; bee bread, n = 17; and royal jelly, n = 15). All the samples were tested for the presence of six of the most common honey bee-associated viruses—Deformed wing virus (DWV), Acute bee paralysis virus (ABPV), Chronic bee paralysis virus (CBPV), Sacbrood virus (SBV), Kashmir bee virus (KBV), and Black queen cell virus (BQCV)—using a reverse transcription polymerase chain reaction (RT-PCR). We successfully detected six records of DWV (10.5%, 6/57), four of ABPV (7.0%, 4/57), three of Israeli acute paralysis virus (IAPV) (5.3%, 3/57), and two of BQCV (3.5%, 2/57). Using ABPV primers, we also successfully detected the presence of IAPV. The obtained viral sequences were analyzed for phylogenetic relationships with the highly similar sequences (megablast) available in the GenBank database. The Bulgarian DWV isolates revealed a high homology level with strains from Syria and Turkey. Moreover, we successfully detected a DWV strain B for the first time in Bulgaria. In contrast to DWV, the ABPV isolates formed a separate clade in the phylogenetic tree. BQCV was closely grouped with Russian isolates, while Bulgarian IAPV formed its own clade and included a strain from China. In conclusion, the present study demonstrated that eRNA can be successfully used for molecular detection of honey bee-associated viruses in bee products. The method can assist the monitoring of the health status of honey bee colonies at the local, regional, and even national levels.

## 1. Introduction

In recent years, the use of environmental DNA (eDNA) as a molecular approach to species identification and distribution, population dynamics, and biodiversity of different taxa has gained much popularity [1,2,3,4]. Since the honey bee comes into contact with various environmental sources (soil, water, and air), the use of eDNA has been successfully implemented in beekeeping as well [5,6]. One of the most common applications of eDNA or environmental RNA (eRNA) in beekeeping is for the purpose of molecular identification of various pathogens, parasites, and pests in bee colonies. Pooled honey samples are most often used as a research approach [7,8,9]. For example, Ribani et al. [10] used eDNA for the molecular detection of nine honey bee pathogens and parasites (*Paenibacillus larvae*, *Melissococcus plutonius*, *Nosema* spp., *Ascosphaera apis*, *Lotmaria passim*, *Acarapis woodi*, *Varroa destructor*, and *Tropilaelaps* spp.). Molecular analysis of eDNA was also used for assessing the presence of honey bee pests: the small hive beetle (*Aethina tumida*) and the greater wax moth (*Galleria mellonella*) [11]. Molecular analysis of non-conventional samples (e.g., honey) is increasingly used in practice for the assessment, detection, and determination of the distribution of a large number of pathogens and pests in bee colonies; it provides useful information regarding the conduct of epidemiological analyses and for monitoring purposes.

Other bee products, such as pollen [12,13,14], wax [12], and royal jelly [15,16], are less frequently used. The successful identification of different pathogens, parasites, and pests largely depends on the target bee product. For example, because the average pH of honey is 3.9 (with a typical range of 3.4 to 6.1) [17], the extracted DNA is highly degraded, and the amplification of PCR products larger than 200 bp is almost impossible [10]. A similar pH value is observed in royal jelly (3.6 to 4.2) [18].

Some of the most significant threats regarding honey bee colonies’ health are honey bee-associated viruses [19,20,21,22]. So far, there are more than 24 identified honey bee viruses [22], which annually cause heavy losses in beekeeping [23,24,25]. The most common honey bee-associated viruses are presented in Table 1. Usually, the majority of them have positive-sense single-stranded RNA genomes, and many belong to the order Picornavirales [26]. The viruses of the family Dicistroviridae account for the largest proportion among honey bee-associated viruses. This family includes the most common bee viruses—Israeli acute paralysis virus (IAPV), Kashmir bee virus (KBV), Acute bee paralysis virus (ABPV), and Black queen cell virus (BQCV) [27,28]. The family Iflaviridae represents another well-known honey bee-associated virus family, which includes the Deformed wing virus (DWV), the Sacbrood virus (SBV), and the Slow bee paralysis virus (SBPV) [27,29]. There are three main DWV genotypes—DWV type A [30], DWV genotype B [31], and type C, which was found to be 98.2% identical to the DWV type A [32].

Conventional approaches for the molecular detection of honey bee-associated viruses involve the use of forager bees, pupae, larvae, and, less often, eggs, drone semen, and feces as biological material [33]. Alternatively, the viral RNA may be detected in different honey bee products—honey, pollen, and bee bread [12,13,34,35]. This represents an innovative, non-invasive, and relatively fast approach for the molecular detection of the most widespread bee viruses. So far, there have not been many reports on the use of bee products as a source for the detection and identification of viral infections in the honey bee.

Therefore, the aim of the present research is to carry out the molecular detection of bee viruses in different bee products (pollen, bee bread, and royal jelly) with a view to monitoring the spread of viral pathogens in different regions of Bulgaria.

**Table 1 vetsci-11-00369-t001:** The most common viruses detected in managed honey bee *A. mellifera*. Viruses with partial genome sequences are marked with an asterisk (*); ^a^ viruses detected mostly in *V. destructor*; n.a.—“not available”; ?, unknown taxonomic status.

Genome Type	Order	Family	Virus	Acronym	Ref. Sequence Acc. No.	References
ssRNA(+)	Picornavirales	Dicistroviridae	Acute bee paralysis virus	ABPV	NC 002548	Govan et al. [36]
				KBV	NC 004807	de Miranda et al. [37]
				IAPV	NC 009025	Maori et al. [38]
			Black queen cell virus	BQCV	NC 003784	Leat et al. [39]
ssRNA(+)	Picornavirales	Iflaviridae	Deformed wing virus	DWV-A	NC 004830	Lanzi et al. [40]
				DWV-B	NC 006494	Ongus et al. [31]
			Egypt bee virus	DWV-C	ERS 657948	Mordecai et al. [32]
				DWV-D	MT 504363	de Miranda et al. [41]
			Moku virus	MV	KU 645789	Mordecai et al. [42]
			Sacbrood virus	SBV	NC 002066	Ghosh et al. [43]
			Slow bee paralysis virus	SBPV	NC 014137	de Miranda et al. [44]
ssRNA(+)	Tymovirales	Tymoviridae	Bee Macula-like virus	BeeMLV	KT 162924	de Miranda et al. [45]
	?	?	Cloudy wing virus *	CWV	n.a.	-
	?	?	Chronic bee paralysis virus	CBPV	NC 010711	Olivier et al. [46]
ssRNA(+)	Nodamuvirales	Sinhaliviridae	Lake Sinai virus	LSV-1	HQ 871931	Runckel et al. [47]
				LSV-2	HQ 888865	
				LSV-3	MH 267700	Thaduri et al. [48]
				LSV-4	KM 886903	Ravoet et al. [49]
dsDNA	Megavirales	Baculoviridae	*Apis mellifera* filamentous virus ^a^	AmFV	MH 243376	Gauthier et al. [50]

## 2. Materials and Methods

### 2.1. Ethical Statement

All the experiments were conducted in accordance with the local ethical committee laws and regulations regarding the European honey bee (*Apis mellifera*), which is neither an endangered nor a protected species.

### 2.2. Geographic Distribution of the Collected Samples

A total of 57 samples from different bee products (pollen, n = 25; bee bread, n = 17; royal jelly, n = 15) were collected in the period 2020–2023 from different regions in Bulgaria (Figure 1). The pollen samples were obtained by pollen traps placed at the entrance of the hive. Briefly, foragers carrying pollen were forced to pass through a pollen trap consisting of a grid with precisely calibrated holes. Pollen was collected daily in humid weather and less frequently in dry weather. In order to avoid deterioration of its quality and the appearance of bacteria and molds, the harvested pollen was dried in special dryers, or simply in a warm and dry place, at a temperature no higher than 40 °C. The bee bread was harvested directly from the honeycombs. All the bee bread samples were stored at −20 °C until evaluation. Finally, the royal jelly was obtained manually by using a silicone spatula 3 days after the emergence of queen cells. The royal jelly was stored at −18 °C to −20 °C prior to molecular analysis. The samples (except bee bread) were not from a single hive; instead, they were pooled as they were obtained by routine procedures for their collection. The pollen samples were either from beekeepers or purchased from the trade market.

### 2.3. Gene Selection

Gene selection was carried out based on available viral sequences in the GenBank database (https://www.ncbi.nlm.nih.gov/genbank/ accessed on 22 February 2023). Regarding DWV and SBV, we amplified the part of the RNA-dependent RNA polymerase gene (RdRp) situated between positions 8556 bp and 8960 bp and between 7747 bp and 8172 bp according to the reference genome Acc. Nos. NC 004830 and NC 002066, respectively [40,43]. These regions were amplified with primer sets described by Stoltz et al. [51] and Tentcheva et al. [52], respectively. A fragment from the 5′-proximal Open Reading Frame (ORF 1) was chosen for the molecular detection of BQCV. This fragment covered the 4611 bp–5034 bp positions from the reference viral genome NC 003784 [39] and was amplified with primers recommended by Tentcheva et al. [52]. Regarding ABPV, a fragment (coding part of the replicase polyprotein) situated between positions 5271 bp and 5722 bp (Ref. seq. NC 002548) [36] was chosen for amplification with primers designed by Tentcheva et al. [52]. For the molecular detection of CBPV, we investigated the region covering 2580 bp to 3034 bp (part of the RdRp gene) according to the reference sequence NC 010711 (Chronic bee paralysis virus RNA 1) [46]. The primer set was according to Ribière et al. [53]. Finally, the identification of KBV was carried out by amplification of the part of the region encompassing 5406 bp–5800 bp (part of the non-structural polyprotein) according to the reference sequence NC 004807 [37] and was identified by primers reported by the same authors [37].

### 2.4. Total RNA Extraction, Copy DNA (cDNA) Synthesis, and RT-PCR Amplification

Total RNA isolation was carried out by using a commercial GeneMATRIX Universal RNA Purification Kit (Cat. No. E3598, EURx Ltd., Gdansk, Poland) according to the manufacturer’s instructions. We used a protocol for plant tissue RNA purification of samples rich in polysaccharides. This protocol includes an LG buffer containing Cetyltrimethylammonium bromide (CTAB), which binds to the polysaccharides when the salt concentration is high, thus precipitating polysaccharides in the solution.

We performed copy DNA (cDNA) synthesis by utilizing a smART First Strand cDNA Synthesis Kit (Cat. No. E0804, EURx Ltd., Gdansk, Poland) according to the manufacturer’s instructions. The reaction mixture contained 14 µL of purified RNA, 4 µL of 5 × NG cDNA Buffer, 1 µL NG dART RT Mix, and 1 µL random hexamer primers in a total volume of 20 µL. The cDNA was synthesized in a thermocycler (BIOER Technology Co., Ltd., Kampenhout, Belgium) under the following conditions: 25 °C for 10 min, followed by 50 min at 50 °C; finally, the reaction mixtures were terminated by incubating at 85 °C for 5 min. The obtained cDNA was stored at −20 °C until analysis.

The RT-PCR reactions were performed with the investigated viral primer sets. The RT-PCR mixtures contained 25 µL of tiOptiTaq PCR Master Mix (2x) (Cat. No. 2726, EURx Ltd., Gdansk, Poland), 1 µM of each virus-specific primer (FOR/REV), and 5 µL of template cDNA for a total volume of 50 µL. The thermal cycling conditions were as follows: initial denaturation at 94 °C for 5 min; 30 cycles (denaturation at 94 °C for 30 s; primer annealing at 50 °C for 30 s; extension at 72 °C for 1 min); and final extension at 72 °C for 10 min. Negative and positive controls were included in each run of the RT-PCR reaction. We used as positive controls RNA samples that were extracted from honey bees and confirmed to be virus-positive by a previous RT-PCR assay [54].

The amplified products were checked on 1% agarose gel electrophoresis stained with SimplySafe™ (Cat. No. E4600, EURx Ltd., Gdansk, Poland). The fragment size was determined by a MassRuler Low Range DNA Ladder (Cat. No. SM0383, Thermo Fisher Scientific Inc., Waltham, MA, USA). The obtained RT-PCR products were purified with a GeneMATRIX Short DNA Clean-Up Purification Kit (Cat. No. E3515, EURx Ltd., Gdansk, Poland) and sequenced in both directions by a PlateSeq kit (Eurofins Genomics Ebersberg, Gdansk, Germany).

### 2.5. Bioinformatics and Molecular Phylogenetic Analysis of Honey Bee-Associated Viruses

The obtained DNA sequences were manually edited and aligned by using MUSCLE software [55] in the MEGA v. 11.0.13 program [56]. All alignments included complete viral genome reference sequences—DWV, Acc. No. NC 004830 [40]; SBV, Acc. No. NC 002066 [43]; BQCV, Acc. No. NC 003784 [39]; ABPV, Acc. No. NC 002548 [36]; CBPV, Acc. No. NC 010711 [46]; and KBV, Acc. No. NC 004807 [37]. The generated sequences from the present study—DWV (382 bp), ABPV (431 bp), BQCV (420 bp), and IAPV (445 bp)—were deposited in the GenBank database (https://www.ncbi.nlm.nih.gov/ accessed on 13 May 2024) under Acc. Nos. PP719389—PP719403. To evaluate the phylogenetic relationships between Bulgarian sequences and the most similar sequences of other countries’ viral isolates (obtained from GenBank), we performed phylogenetic analysis using the MEGA v. 11.0.13 program [56]. The phylogenetic tree was constructed using the maximum likelihood method and a bootstrap value of 1000 replicates for each investigated virus. For detecting potential mutations in viral isolates from the current study, we additionally included sequences obtained in previous studies [12,54] in the phylogenetic analysis. We also performed a one-way ANOVA test, using F distribution with a post hoc Tukey HSD test to compare the frequency of positive and negative virus samples between different regions of the country.

## 3. Results

### 3.1. Prevalence of Detected Honey Bee-Associated Viruses in Different Regions of Bulgaria

We tested different bee products—pollen, n = 25; bee bread, n = 17; and royal jelly, n = 15—for the most distributed honey bee viruses. DWV had the highest prevalence (10.5%, 6/57) (Figure 2). The highest incidence of this virus was detected in the central part of the country. Also, DWV was detected in all the investigated bee products. ABPV had the second highest occurrence, with a relatively low frequency (7.0%, 4/57). In contrast to DWV, ABPV was concentrated mostly in the northern part of the country. All the positive cases of that virus were detected only in royal jelly. Interestingly, the molecular detection of IAPV was conducted with a primer set for IAPV. However, this scenario was not unusual considering the fact that the primers cover the part of the replication region of the viral genome for both viruses, where the viral sequences are closely related between ABPV and IAPV. Like DWV, IAPV was detected mostly in the central part of the country, with a frequency of 5.3% (3/57). Because of the high genetic similarity of non-structural protein sequences between the two viruses, IAPV was detected only in royal jelly. Finally, BQCV was established with the lowest sequences—only two positive samples were found in the central and the northern part of the country. Moreover, we detected instances of co-infection in some bee products—BQCV, ABPV, and IABV in royal jelly from southeast Bulgaria as well as BQCV and ABPV again in royal jelly from northeast Bulgaria. Surprisingly, we were not able to detect SBV, which has been identified with a relatively high frequency in our previous study [12,54].

When comparing positive and negative virus samples from three regions of the country—central, northern, and eastern Bulgaria—no significant differences were observed (ANOVA with Tukey’s HSD post hoc test, F =  0.5625, df = 2, *p* = 0.5841). Statistically insignificant differences were detected between central and northern as well as between central and eastern Bulgaria (*p* = 1.0 and 0.6394, respectively).

### 3.2. Molecular Phylogenetic Relationships of Honey Bee-Associated Viruses

We explored the phylogenetic relationship of Bulgarian DWV isolates with genotypes from different countries, based on the highly similar sequences (megablast) available in the GenBank database, according to a sequence analysis of the part of the RdRp gene of the viral genome (382 bp) (Figure 3).

The constructed cladogram showed that some viral strains tended to form their own branches—Asian, European, North (USA) and South American (Brazilian) clades. This demonstrated the geographic separation of DWV genotypes. With few exceptions, the phylogenetic analysis of the Bulgarian isolates formed a separate clade (Bulgarian clade 2) (Figure 3). This branch also included DWV representatives from our previous study [12] as well as isolates from Turkey and Syria. The remaining four DWV isolates were dispersed in the phylogenetic tree, which suggests multiple origins of the Bulgarian population. Interestingly, we found DWV genotype B, which is the first report for the occurrence of this type in Bulgaria.

To analyze the phylogeny of the Bulgarian ABPV isolates, we sequenced a part of the replicase polyprotein coding sequence (431 bp) relative to the reference viral genome (Acc. No. NC 002548). Figure 4 demonstrates that the Bulgarian representatives clustered close to each other, which indicated the monophyletic origin (Bulgarian clade) with a relatively homogenous population of ABPV in the country. Similarly, Swiss and Chinese isolates also formed independent clades (Swiss and Chinese clades). The Slovenian representatives were dispersed throughout the phylogenetic tree, demonstrating the presence of quite heterogeneous populations of ABPV with multiple origins in Slovenia. Regarding the phylogeny of the Bulgarian ABPV isolates from *Apis mellifera* [54], it should be noted that they separated from that of the current study, showing that a mutation possibly occurred in the viral genome.

The phylogeny of IAPV was inferred on the basis of a 445 bp fragment covering the part of the replicase-associated protein region according to the reference viral genome (Acc. No. NC 009025). In a similar manner to the phylogeny of DWV, the IAPV isolates clearly demonstrated geographic separation and formed five clades (Bulgarian, Chinese, South Korean, Australian, and USA) (Figure 5). The Chinese, Australian, and USA populations showed monophyletic clustering, suggesting the existence of homogenous isolates in these countries. The Bulgarian population formed a separate clade that includes a representative of China (Acc. No. MZ 821843). The monophyletic clustering of Bulgarian genotypes along with the Chinese isolate suggests that they share a common origin and that relatively homogenous populations of IAPV exist in Bulgaria.

To infer the phylogeny of two Bulgarian BCQV isolates, we determined the 420 bp fragment encoding the nonstructural polyprotein (ORF 1); then, a phylogenetic tree was constructed with those of the other most similar isolates available in GenBank (Figure 6). The Bulgarian isolates were grouped close together with the Russian isolates. In contrast, the phylogenetic position of the Bulgarian isolate from *Apis mellifera* [54] demonstrated different clustering, suggesting that genetic distance exists between BQCV genotypes isolated from bee products and honey bees. The Polish, UK, Swedish, and Chinese representatives form separate clusters, suggesting low genetic diversity of the BQCV isolates from these countries. Slightly different is the phylogeny of the Australian clade, as it includes one isolate from South Korea (Acc. No. JX 149531). This indicates that these isolates share a common origin.

## 4. Discussion

The current study represents a non-invasive approach for molecular detection of honey bee-associated viruses in different bee products (bee pollen, bee bread, and royal jelly) on the basis of eRNA analysis. Thus, there is no need to catch forager bees from every hive, avoiding the difficulty of crushing samples during the homogenization process, as well as the risk to human health when collecting sick or dead bees. Therefore, it accelerates the early detection and timely eradication or control of diseases. Moreover, in some cases, by applying simple technological measures, we can prevent the appearance of clinical signs/forms of the disease and thus reduce harmful economic losses for beekeepers. For this reason, the early diagnosis of subclinical levels of pathogenic microorganisms in the hive or apiary is a priority for effective disease prevention in beekeeping practices.

The obtained results show that all the bee products are suitable biological products for identification and control of the dispersal of the investigated viruses. As far as we know, this was the first study using royal jelly as a source for honey bee virus detection. One of the major limitations of molecular analysis of eRNA is that compared to eDNA, eRNK degrades more rapidly in biological samples [58]. On the other hand, the susceptibility of RNA to degradation makes it difficult to work with. The collection of RNA samples requires dedicated sampling protocols, more careful preservation, and storage. There is also additional processing time, as well as costs, associated with the isolation and reverse transcription of RNA [59], making it more expensive and challenging and thus a less attractive molecule to work with.

One of the main questions that arises in the present study is related to how degraded the RNA in honey bee products could be while still remaining suitable for molecular analysis. Indeed, all bee products used in the current study had a low pH—pollen (3.8 to 6.3) [60], royal jelly (3.6 to 4.2) [18], and bee bread (3.8–4.3) [61]. A possible explanation for RNA stability in an acidic environment is that viral capsid proteins protect the RNA from degradation, keeping it intact and suitable for molecular analysis. Another scenario is associated with the hypothesis that the acidic pH environment stabilizes key intra- and intermolecular RNA bonds, the RNA phosphodiester bond, and the aminoacyl-(t) RNA and peptide bonds [62].

It is known that honey bee-associated viruses cause great losses in the beekeeping sector annually [21,63,64,65]. Assessing the health status of bee populations is an essential component of management measures to ensure their survival and resilience, especially for those affected by pathogens. Environmental RNA could provide an insight into pathogenic infections and inform managers that honey bee populations are physiologically responding to viral pathogens [66]. The samples used in the present study are mostly pooled, which increases the success rate regarding not only the detection of honey bee-associated viruses but also the identification of other pathogens and pests [8,17]. The advantage of using such an approach is the obtainment of information about the epizootic situation at the local, regional, and even national levels. It is essential to protect healthy bee colonies when purchasing broods, hives, or queen bees. What the health risk is when feeding healthy bee colonies with honeycombs or bee products from affected ones remains an open question.

As shown in Figure 2, the majority of virus-positive samples were detected in central Bulgaria. This is not surprising, considering the fact that a large number of tourist sites and attractions are concentrated in this part of the country. The year-round movement of people and goods in this location is a prerequisite for the transfer of bee viruses from other regions of the country. In contrast, the lowest virus prevalence was observed in northwest Bulgaria. The latter is the fastest depopulating and the least economically developed region in the entire country. Therefore, as could be expected, the highest percentage of negative samples was found in this part of Bulgaria.

We performed a phylogenetic analysis of each detected honey bee-associated virus to reveal the phylogenetic position of Bulgarian isolates and the most homologous sequences available in GenBank. Additionally, we included the isolates from our previous investigation [12,54] in the phylogeny to test whether there were any changes in the homogeneity or heterogeneity of the viral populations.

The sequence analysis of DWV showed that almost all of them belonged to the DWV type A (Figure 3). The only exception was the isolate Acc. No. PP719398, which was assigned to the DWV type B. So far, this was the first report for the circulation of the DWV type B in Bulgarian apiaries. Interestingly, the DWV type B was detected simultaneously with the DWV type A in the same pollen sample. The majority of the DWV type A were grouped closely to the Bulgarian genotypes obtained from our previous study [12], as well as one isolate from Turkey (Acc. No. KU5217779). These results demonstrated that the phylogenetic position of the DWV type A from our two investigations did not show different clustering. The remaining viral isolates were dispersed in the phylogeny, suggesting that DWV possessed multiple origins and that the Bulgarian populations are heterogeneous.

The phylogenetic analysis of ABPV showed that all isolates from the present study formed a separate branch, which confirmed the monophyletic clustering of Bulgarian representatives. This demonstrated that the population of Bulgarian ABPV is homogenous and geographically separated (Figure 4). The phylogenetic position of ABPV from our previous study [54] was retained, since Bulgarian isolates were grouped along with other Central European genotypes from Slovenia and the Czech Republic. The different clustering of ABPV isolated from the honey bees and investigated bee products may be associated with a difference in the collection of data and/or some degree of virus diversity.

An interesting finding of this study was the molecular identification of IAPV with a primer set for detection of ABPV. It is well known that both viruses, along with KBV, are part of a complex of closely related viruses from the Family Dicistroviridae [35,67,68]. This primer set amplifies a 452 bp fragment of the RdRp nucleotide sequence (Ref. seq. NC002548) [36]. Figure 7 demonstrates the alignment of this region with the RdRp region of the newly found IAPV.

The forward primer showed the covered region of ABPV, which differed with five substitutions from the sequence of IAPV. The reverse primer showed only two mutations between the two viruses. According to Maori at al. [38], this primer set was not specific for ABPV but had the potential to amplify the sequence from the IAPV genome. Due to genetic similarity, the potential for cross-amplification with related viruses became clear with the subsequent sequencing of the ABPV, KBV, and IAPV genomes [37,69]. Another issue arising with regard to discriminating between ABPV and IAP is associated with the available sequences in the GenBank database. Many of the IAPV isolates, particularly those from France, Russia, and Australia, date from before 2004 (when the first IAPV sequences were made available). Hence, while duly classified as KBV according to the information available at the time [38], these isolates still remain misclassified, since it is clear now that they belong to a different taxon.

The phylogenetic analysis of Bulgarian IAPV strains showed that all of them formed a separate branch that included an isolate from China. This suggests that the Bulgarian isolates were geographically separated and appeared to be a homogenous population. This was the first detection of IAPV in Bulgaria, and we hope that additional data will contribute to elucidating the molecular phylogeny of IAPV.

Notably, we found for the first time the presence of IAPV not only in honey bee samples but also in bee products (royal jelly). This gives us reason to believe that molecular analysis of eRNA from bee products would provide beekeepers with an opportunity for more comprehensive monitoring of bees’ health status. Therefore, programs of bee disease screening incorporating this innovative approach should be included in regular control procedures.

Finally, the phylogeny of BQCV was constructed based on the sequence analysis of the fragment cover part of ORF 1. The Bulgarian isolates shared a common branch with Russian strains (Figure 6). This suggests that BQCV isolates from the two countries have a common origin. This is not unusual, considering the trade relationship and exchange of goods between the two countries. Moreover, the main vector of distribution of this virus—the ectoparasitic mite *Varroa destructor* was introduced for the first time in Europe from the former USSR to Bulgaria in 1970 [70]. Presumably, this occurred via the introduction of honey bees and queen bees from Russia to improve the productivity of the Bulgarian native honey bee [70]. In contrast to the phylogenetic position of BQCV detected in bee products, the BQCV strain isolated from *Apis mellifera* [54] showed a different position on the phylogenetic tree. This isolate clustered between the Swedish and the Chinese clades, which confirmed that its phylogenetic position remained unchanged compared to our previous study [54]. Molecular detection of the six most common bee viruses (BQCV, Kashmir bee virus—KBV, DWV, ABPV, SBV, and CBPV) in honey samples from Serbia (30 samples originating from 12 different regions in central and northern Serbia and 5 imported), successfully detected only two of them—BQCV and KBV [34]. In contrast to our results, the phylogenetic analysis of BQCV showed two different genotypes. The Serbian isolates from the first one were the closest to the Hungarian isolate and included a sublineage clustered together with a Chinese isolate, while the second genotype included only one sample from the Sabac region in Serbia.

## 5. Conclusions

This study demonstrates the use of a novel, non-invasive approach (eRNA) for molecular detection of honey bee-associated viruses in different bee products (pollen, bee bread, and royal jelly). Since the examined samples are most often pooled, this allows monitoring and prevention of viral diseases not only locally, but also at the regional and national levels. A large part of viral diseases are most often asymptomatic, and their identification in bee products by molecular analysis of eRNA is important for determining the epidemiological data and the distribution of different viral diseases in the country.

The phylogenetic analysis of the detected honey bee-associated viruses showed that DWV revealed the highest genetic diversity and a heterogeneous population. In contrast, ABPV and IAPV formed separate clades, suggesting the monophyletic origin and homogeneous populations of these strains. The phylogenetic position of BQCV was completely different from that of the other viruses, since the Bulgarian isolates grouped together with Russian strains, suggesting the common origin of BQCV from the two countries.

Additional investigations would be needed to resolve the question of whether viral particles in bee products remain invasive over time and pose a potential danger of infecting healthy bee colonies. Considering that honey bee health threats cannot be regarded as local problems, the results of the present study may be used as a starting point for further molecular detection and for determining the dissemination, co-occurrence, and prevalence of other pathogens, parasites, and pests, based on honey bee products.

## Figures and Tables

**Figure 1 vetsci-11-00369-f001:**
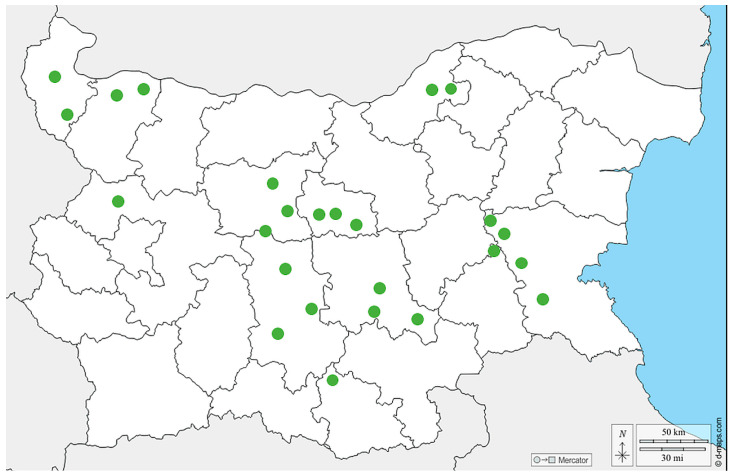
Map of the sampling sites from different regions in Bulgaria.

**Figure 2 vetsci-11-00369-f002:**
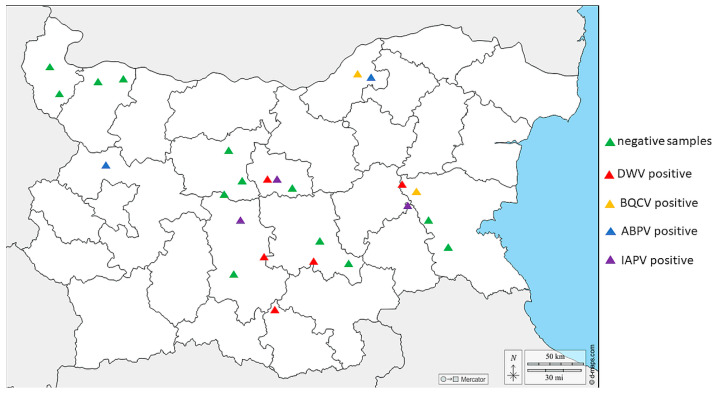
Map presenting the distribution of the examined positive/negative samples of honey bee-associated viruses in Bulgaria.

**Figure 3 vetsci-11-00369-f003:**
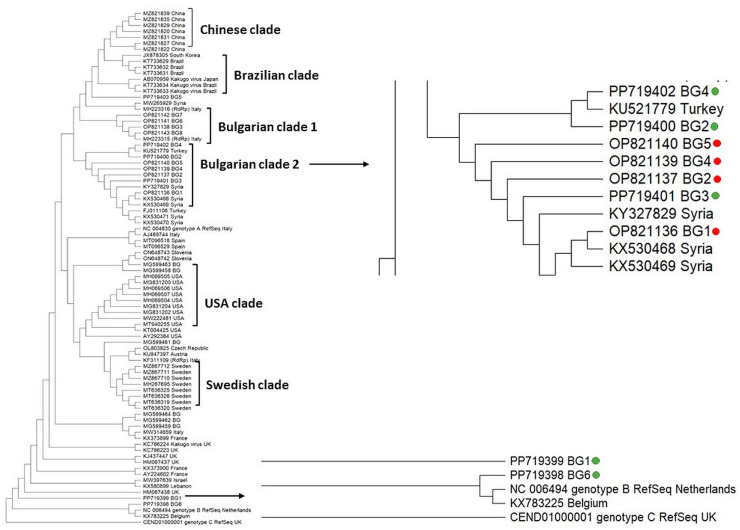
Phylogeny of Deformed wing virus (DWV) isolates from Bulgaria and other countries. The phylogenetic tree based on alignment of the fragment of RdRp gene sequences of DWV isolates from different countries was inferred by using the maximum likelihood method and the Tamura 3-parameter model and then selecting the topology with a superior log likelihood value [57]. The indicated branching topology was evaluated by bootstrap resampling of the sequences of 10,000 replicates. Each isolate is indicated by country of isolation and GenBank accession number. Bulgarian isolates identified by this study are presented in green and red [12].

**Figure 4 vetsci-11-00369-f004:**
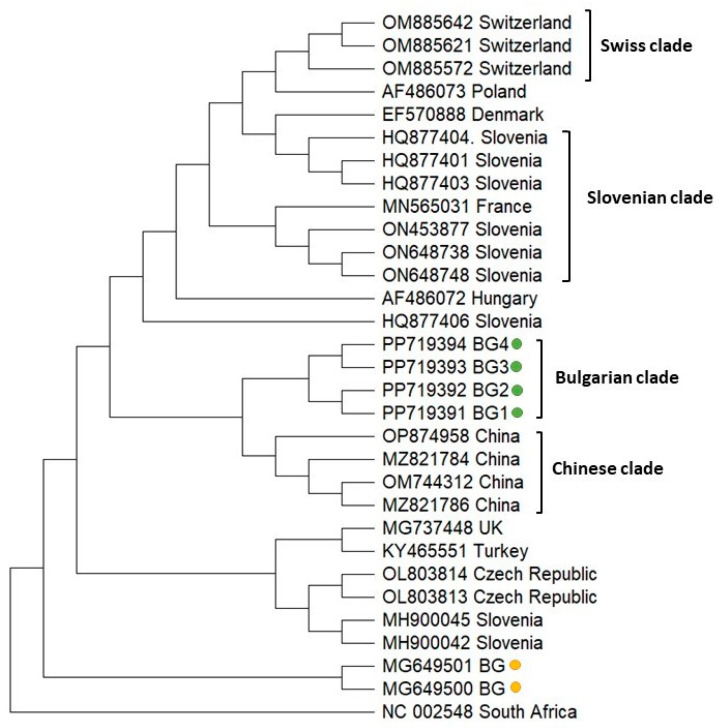
Phylogeny of Acute bee paralysis virus (ABPV) isolates from Bulgaria and other countries. The phylogenetic tree based on alignment of the part of the replicase polyprotein gene sequences of ABPV isolates (431 bp) from different countries was inferred by using the maximum likelihood method and the Tamura 3-parameter model and then selecting the topology with a superior log likelihood value [57]. The indicated branching topology was evaluated by bootstrap resampling of the sequences of 10 000 replicates. Each isolate is indicated by country of isolation and GenBank accession number. Bulgarian isolates identified by this study are presented in green and yellow [54].

**Figure 5 vetsci-11-00369-f005:**
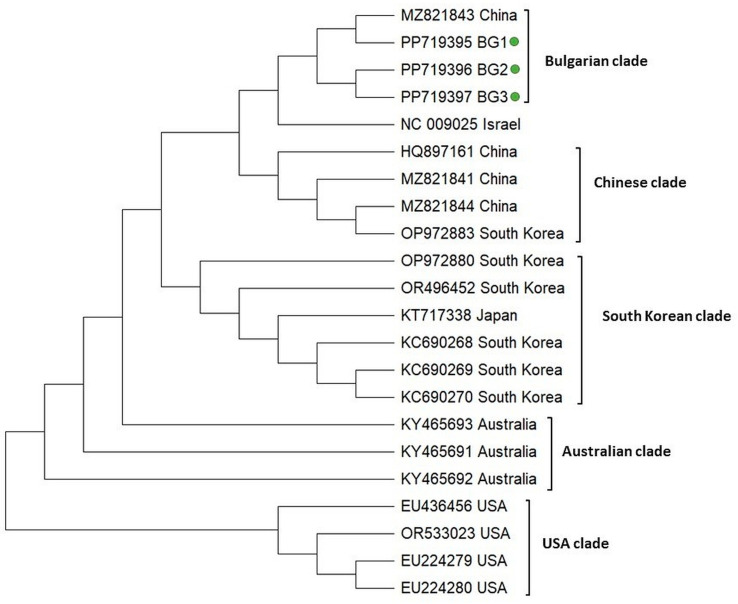
Phylogeny of Acute bee paralysis virus (IAPV) isolates from Bulgaria and other countries. The phylogenetic tree based on alignment of the part of the replicase polyprotein gene sequences of IAPV isolates (445 bp) from different countries was inferred by using the maximum likelihood method and the Tamura 3-parameter model and then selecting the topology with a superior log likelihood value [57]. The indicated branching topology was evaluated by bootstrap resampling of the sequences of 10 000 replicates. Each isolate is indicated by country of isolation and GenBank accession number. Bulgarian isolates identified by this study are presented in green.

**Figure 6 vetsci-11-00369-f006:**
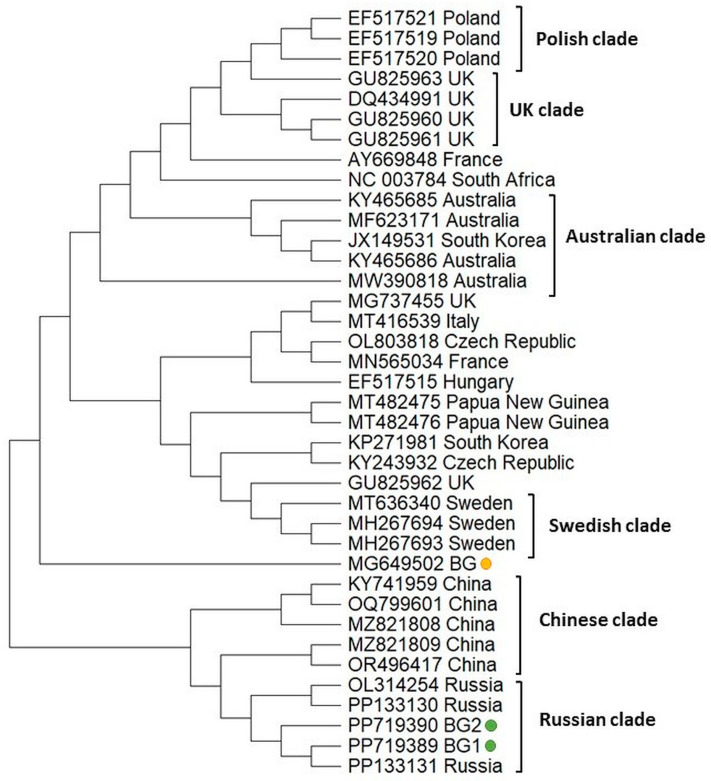
Phylogeny of Black queen cell virus (BQCV) isolates from Bulgaria and other countries. The phylogenetic tree based on alignment of the part of the ORF 1 gene sequences of BQCV isolates (420 bp) from different countries was inferred by using the maximum likelihood method and the Tamura 3-parameter model and then selecting the topology with a superior log likelihood value [57]. The indicated branching topology was evaluated by bootstrap resampling of the sequences of 10,000 replicates. Each isolate is indicated by country of isolation and GenBank accession number. Bulgarian isolates identified by this study are presented in green and yellow.

**Figure 7 vetsci-11-00369-f007:**
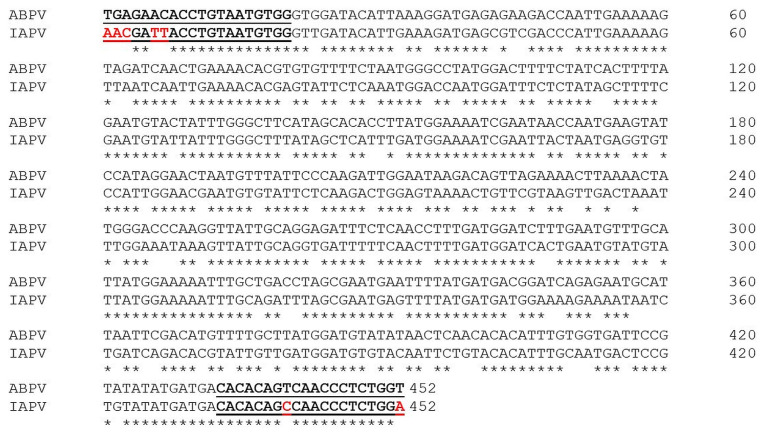
ABPV–IAPV nucleotide alignment of a 452 bp fragment of the RdRp gene. Asterisks (*) indicate identical nucleotide positions in the alignment. The reported sequences are part of reference viral genomes (ABPV Acc. No. NC 002548; IAPV Acc. No. NC 009025). The PCR primer regions are underlined and in bold.

## Data Availability

All data are available upon request.

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
