# Peer review of "Molecular Detection and Phylogenetic Relationships of Honey Bee-Associated Viruses in Bee Products"

_vetsci, 2024, doi:10.3390/vetsci11080369_

Round 1

Reviewer 1 Report

Comments and Suggestions for Authors

Review grammar throughout the text.

The authors could add a little more literature on the use of eDNA/eRNA in detecting pathogens in beekeeping or other animal-derived products and how its use and practice impact these products.

The authors should include greater detail on sampling procedures.

Expand the discussion to include a comparison of previous studies and the implications of the findings for theory and practice. Address possible limitations of the study and suggest future research.

Comments on the Quality of English Language

Review grammar throughout the text.

Author Response

Dear Reviewer

Thank you for dedicating your time and effort to review our manuscript. We have carefully revised the manuscript according to your comments and suggestions. Please find the point to-point responses below and the corresponding revisions in the re-submitted files. The line numbers mentioned in the responses refer to the lines in the highlighted revised version. All changes were made using the Track Changes mode of MS Word.

Review grammar throughout the text.

Response: We have checked the English again.

The authors could add a little more literature on the use of eDNA/eRNA in detecting pathogens in beekeeping or other animal-derived products and how its use and practice impact these products.

Response: We agree and we have provided additional information according to the reviewer’s recommendations (Line 57-66).

The authors should include greater detail on sampling procedures.

Response: We agree and we have provided greater detail on sampling procedures (Line 112-121).

Expand the discussion to include a comparison of previous studies and the implications of the findings for theory and practice. Address possible limitations of the study and suggest future research.

Response: In the available literature, there is only one study on molecular detection of bee viruses in bee products (honey). For this reason, we have included only the results of this study (Line 435-442). We have also added information about the implications of molecular analysis of eRNA for beekeeping practice (Line 414-421). We have addressed possible limitations (Line 322-328) and future research (Line 462-465).

Review grammar throughout the text.

Response: We have reviewed the entire text grammatically.

Reviewer 2 Report

Comments and Suggestions for Authors

I thank the journal editor for the opportunity to review this article.

Comments:
A table of common viruses affecting honey and honey products would make the Introduction section more readable.
Materials and Methods:
It would be better if Figure 1 only shows the geographical distribution of the samples and another figure like Figure 1, along with the number of positive/negative samples, is created and moved to the Results section.
A picture showing the flow of the experimental procedure would also be helpful.
The geographical area should be commented on in the discussion section, and the different presence of viruses should be justified. The part of the analysis and the presentation of the results concern more methodological problems of molecular analysis. This is also documented by the fact that there is no statistical analysis of the comparison of the results between the regions, e.g. Chi-square test, Krusal-Wallis test or ANOVA.

Author Response

Dear Reviewer

Thank you for dedicating your time and effort to review our manuscript. We have carefully revised the manuscript according to your comments and suggestions. Please find the point to-point responses below and the corresponding revisions in the re-submitted files. The line numbers mentioned in the responses refer to the lines in the highlighted revised version. All changes were made using the Track Changes mode of MS Word.

Comments:

A table of common viruses affecting honey and honey products would make the Introduction section more readable.

Response: We agree and we have prepared a table of common viruses affecting honey and honey products.

Materials and Methods:

It would be better if Figure 1 only shows the geographical distribution of the samples and another figure like Figure 1, along with the number of positive/negative samples, is created and moved to the Results section.

Response: We agree and we have prepared a figure with the geographical distribution of the samples.

A picture showing the flow of the experimental procedure would also be helpful.

Response: We agree and we have provided additional information concerning sample collection and processing (Line 112-121).

The geographical area should be commented on in the discussion section, and the different presence of viruses should be justified.

Response: We agree and we have commented on virus prevalence in different regions in the country (Line 351-359).

The part of the analysis and the presentation of the results concern more methodological problems of molecular analysis. This is also documented by the fact that there is no statistical analysis of the comparison of the results between the regions, e.g. Chi-square test, Krusal-Wallis test or ANOVA.

Response: We applied a One Way ANOVA test, using F distribution with a post hoc Tukey HSD test to compare the frequency of positive and negative honey samples between different regions of the country (Line 197-199; 226-230).

Reviewer 3 Report

Comments and Suggestions for Authors

In the manuscript " Molecular detection and phylogenetic relationships of honey bee-associated viruses in bee products”, Salkova et al describes molecular analysis of environmental RNA as a tool for detection of six of the most widespread viruses in honey bee colonies in bee products (pollen, bee bread, and royal jelly) from different regions in Bulgaria. This study represents a non-invasive approach on the basis of eRNA analysis, and thus, there is no need to catch forager bees from every hive, avoiding the difficulty of crushing samples during the homogenization process. This short report is of scientific and practical interest, and relatively well structured and written. It would be considered to check each eRNA stability in different bee products as a normalized control. In addition, catching some bees for assaying the present viruses should be also considered as positive control for identification. The current form of the manuscript is short report suitable to be communication article acceptable for publication in Vet. Sci.

Author Response

Dear Reviewer

Thank you for dedicating your time and effort to review our manuscript.

In the manuscript " Molecular detection and phylogenetic relationships of honey bee-associated viruses in bee products”, Salkova et al. describes molecular analysis of environmental RNA as a tool for detection of six of the most widespread viruses in honey bee colonies in bee products (pollen, bee bread, and royal jelly) from different regions in Bulgaria. This study represents a non-invasive approach on the basis of eRNA analysis, and thus, there is no need to catch forager bees from every hive, avoiding the difficulty of crushing samples during the homogenization process. This short report is of scientific and practical interest, and relatively well structured and written.

It would be considered to check each eRNA stability in different bee products as a normalized control.

Response: Indeed, we have commented on this matter in the Discussion section (Line 329-337). The fact that viral RNA remains intact even after four years of sample storage suggests that most likely the structural proteins of the viral capsid protect it from degradation.

In addition, catching some bees for assaying the present viruses should be also considered as positive control for identification.

Response: For positive controls, we used positive viral samples detected in honey bees in our previous research:

Shumkova R, Neov B, Sirakova D, Georgieva A, Gadjev D, Teofanova D, Radoslavov G, Bouga M, Hristov P. 2018. Molecular detection and phylogenetic assessment of six honeybee viruses in Apis mellifera L. colonies in Bulgaria. PeerJ 6:e5077. https://doi.org/10.7717/peerj.5077

The current form of the manuscript is short report suitable to be communication article acceptable for publication in Vet. Sci.